

# Leaf functional trait variation in a humid temperate forest, and relationships with juvenile tree light requirements

Christopher H. Lusk

Environmental Research Institute, University of Waikato, Hamilton, New Zealand

## ABSTRACT

The species-rich arborescent assemblages of humid tropical forests encompass much of the known range of the leaf economics spectrum, often including >20-fold variation in leaf lifespan. This suite of traits underpins a life-history continuum from fast-growing pioneers to slow-growing shade-tolerant species. Less is known about the range of leaf traits in humid temperate forests, and there are conflicting reports about relationships of these traits with the light requirements of temperate evergreen angiosperms. Here I quantify the range of leaf functional traits in a New Zealand temperate evergreen forest, and relationships of these traits with light requirements of juvenile trees and shrubs. Foliage turnover of saplings of 19 evergreen angiosperms growing beneath gaps (12–29% canopy openness) and in understories (1.2–2.9%) was measured over 12 months. Dry mass per area (LMA), dry matter content, thickness, density and nitrogen content (N) of leaves were also measured. Species minimum light requirements were indexed as the 10th percentile of the distribution of saplings in relation to canopy openness. Interspecific variation of leaf lifespan was ∼6-fold in gaps (0.6 to 3.8 yrs), and ∼11-fold in the understorey (0.7 to 7.7 yrs). Six small tree and shrub species are effectively leaf-exchangers, with leaf lifespans of c.1 year in gaps—albeit usually longer in the shade. Interspecific variation in other leaf traits was 2.5 to 4-fold. Lifespans and LMA of both sun and shade leaves were negatively correlated with species light requirements i.e., positively correlated with shade tolerance. However, light environment (gap vs shade) explained about the same amount of variation in LMA as species' identity did. Species light requirements were not significantly correlated with leaf N, dry matter content, density or thickness—except for a marginally significant correlation with dry matter content of shade leaves. Species light requirements were thus less consistently related to leaf structural traits than appears to be the case in humid tropical forests. Whereas the wide interspecific variation in leaf economic traits of tropical rainforest species outweighs plastic response to light availability, temperate evergreen woody angiosperms appear to occupy a narrower range of the leaf economic spectrum. Standardization of the light environments in which LMA is measured is vital in comparative studies of humid temperate forest evergreens, because of countergradient responses of this trait to light, and because of the relative magnitudes of plastic and interspecific variation in LMA in these forests.

Corresponding author
Christopher H. Lusk,
clusk@waikato.ac.nz

## INTRODUCTION

The unrelenting evergreenness of humid tropical forests belies the vast range of foliage turnover rates revealed by comparative studies of their arborescent assemblages, which often exceeds 20-fold variation (*Reich et al., 1991*; *Russo & Kitajima, 2016*). This wide variation in leaf lifespan, closely linked to a suite of other leaf traits in what has become widely known as the "leaf economics spectrum", underpins a life-history continuum from fast-growing pioneers to slow-growing shade-tolerant species (*Lohbeck et al., 2013*; *Poorter & Bongers, 2006*; *Sterck, Poorter & Schieving, 2006*; *Walters & Reich, 1999*). This continuum is also associated with interspecific variation in wood density (*King et al., 2006*; *Van Gelder, Poorter & Sterck, 2006*). The consistent picture emerging from studies of humid tropical forests is that low leaf mass per area (LMA), high assimilation rates and low wood density enable pioneer trees to rapidly pre-empt gaps and clearings, whereas positive long-term net carbon gain and survival under shade is made possible by robust, long-lived leaves, dense wood, and low respiration rates. The fast-growing pioneers that colonize gaps in humid tropical forests have "high-maintenance" foliage: their low-LMA leaves turn over rapidly (2–6 months) and have high rates of photosynthesis and respiration.

Less can be said with certainty about functional diversity of leaf traits in humid temperate forests, or about trait relationships with species' light requirements and life histories. Tall, fast-growing pioneers of the type found in tropical and subtropical humid forests are known to be lacking from mid-latitude forests (*Lusk, Kooyman & Sendall, 2011a*). Seven-fold variation in leaf lifespans of juvenile trees has been reported from the humid temperate forests of south-central Chile (*Lusk et al., 2008a*), but fewer data are available from comparable assemblages in New Zealand and temperate Australia. In the deciduous angiosperm assemblages typical of continental temperate climates, interspecific differences in leaf lifespan are inevitably muted (*Van Ommen Kloeke et al., 2012*; *Walters & Reich, 1999*). In evergreen temperate forests, leaf lifespan is once again consistently negatively correlated with species' reported light requirements (i.e., positively so with shade tolerance), but there is no such agreement about relationships of leaf structural traits such as LMA with light requirements (*Fajardo & Siefert, 2016*; *Hallik, Niinemets & Wright, 2009*; *Lusk et al., 2011b*; *Lusk & Warton, 2007*).

A lack of standardization of the light environments in which traits are measured might underlie some of the discrepancies in reporting relationships of leaf structural traits with light requirements of temperate forest evergreens (*Keenan & Niinemets, 2016*). LMA shows strong plastic responses to light availability, with sun leaves being much thicker than shade leaves of the same species (*Poorter et al., 2009*), and sometimes also denser. Plastic variation of LMA along light gradients thus runs counter to interspecific variation relating to species' shade tolerance (*Lusk et al., 2010*; *Lusk et al., 2008b*), with the result that sun leaves of light-demanding evergreens can have similar LMA to shade leaves of shade-tolerant species. If juvenile trees are sampled randomly or haphazardly without controlling for light environment, interspecific differences in traits such as LMA may be masked by plastic variation (*Laughlin et al., 2017*), as differential survival along light gradients results in light-demanding species being found on average in better-lit environments than their

more shade-tolerant associates (*Kobe, 1999*; *Lusk, Chazdon & Hofmann, 2006*; *Poorter & Arets, 2003*). The same biases may be present in databases compiled from reviews of the literature (see *Keenan & Niinemets, 2016*).

Here I document the range of leaf traits in a humid temperate forest arborescent assemblage in New Zealand, and relationships of these traits with species' light requirements. Leaf traits were measured in two distinct light environments (understorey shade and treefall gaps), using hemispherical photography to quantify canopy openness. It has recently been shown that the microclimates of clearings at the same site favour very different traits from those of species that regenerate primarily in treefall gaps (*Lusk & Laughlin, 2017*). The present study focuses mainly on species sorting along gap to understorey gradients, rather than the more open environments of clearings.

## MATERIALS AND METHODS

### Study site

The study was carried out in a humid temperate forest in the Lake Okataina Scenic Reserve (38.08°S, 176.42°E), in the North Island of New Zealand. Sampling was carried out in a 300 ha basin lying at about 400m a.s.l. within the reserve, infilled with tephras (mainly rhyolitic) derived from the Okataina Volcanic Centre (*Pullar, Birrell & Heine, 1973*). Climate data from GIS layers indicate a mean annual temperature of 11.7 °C, a frost-free period of 195 days, and mean annual precipitation of 1,659 mm (*IIASA/FAO, 2012*; Landcare Research 2011). Rainfall is evenly-distributed throughout the year. Although juvenile trees growing in clearings are exposed to large vapour pressure deficits (>1.5 kPa) during dry spells, deficits of that magnitude have not been recorded in the understorey and tree-fall gap environments where traits were measured in the present study (*Lusk & Laughlin, 2017*). A research permit to work at the site (66760-RES) was obtained from Department of Conservation.

The disturbance history of the basin has created a complex vegetation mosaic, including a wide range of light environments (*Lusk & Laughlin, 2017*). Most of the basin remains in tall forest with a canopy dominated by *Beilschmiedia tawa* (Lauraceae) up to 30 m tall and scattered emergent conifers up to 45 m, mainly *Dacrydium cupressinum* and *Dacrycarpus dacrydioides* (Podocarpaceae). Conifers occurred at higher densities before selective logging during the mid-20th century (*Nicholls, 1991*); this history of logging has left behind several clearings and a network of skidder tracks, some of which have been converted to walking tracks. At the north end of the basin is a stand of <2 ha with a canopy dominated by *Weinmannia racemosa* (Cunoniaceae), which admits more light to the understorey than the deep-crowned *B. tawa* that predominates elsewhere (*Beveridge, 1973*). As a result, seedlings and saplings of a wide range of species can be found in the understorey of this stand, which presumably owes its origin to a small fire or wind damage. All native tree and shrub species present in the basin are evergreen, except for *Fuchsia excorticata* (Onagraceae) and very occasional *Plagianthus regius* (Malvaceae)—*McGlone et al. (2004)* describe both as deciduous, but report that some northern populations of the former retain some leaves during winter.

## Measurements of leaf traits

I measured a range of leaf traits that have variously been shown to correlate with species' light requirements in other evergreen forests (*Kitajima & Poorter, 2010*; *Lusk et al., 2010*; *Poorter & Bongers, 2006*). Although leaf mass per area (LMA) is the structural trait included in the original leaf economic spectrum concept (*Reich et al., 1991*; *Wright et al., 2004*), it can be informative to partition this trait into leaf thickness and density—it has been reported that density correlates strongly with species' shade tolerance in tropical humid tropical forests, whereas thickness does not (*Kitajima & Poorter, 2010*). Accordingly, leaf density and thickness were measured in addition to LMA. Leaf dry matter content is a widely used alternative to LMA, and has also been found to correlate with species' light requirements in some humid forests (*Lusk et al., 2010*; *Poorter, 2009*).

Two ranges of light environments were defined as sources of sun- and shade-leaf traits (Tables 1, 2). Sun-leaf traits were measured on plants growing beneath tree-fall gaps, or besides roads or walking tracks. Shade-leaf traits were measured on juvenile trees growing mostly in the understorey of the *W. racemosa*-dominant stand at the north end of the basin. In each of these two light environments, five to six juveniles (50–200 cm tall) were selected haphazardly for leaf trait measurements. A Nikon Coolpix 4500 digital camera (Nikon, Tokyo, Japan) and an EC-08 fisheye adaptor were used to take a photo immediately above the apex of each of these selected juveniles (as described above), and Gap Light Analyzer (*Frazer, Canham & Lertzman, 1999*) was used to estimate % canopy openness from each photo. Light environments above plants sampled in gaps ranged from 12.0 to 27.8% canopy openness, compared to 1.2 to 3.0% in the understorey. There was no significant interspecific variation in mean canopy openness above plants sampled in either gaps (ANOVA, $P = 0.97$) or shade ($P = 0.76$). Juveniles of *Coprosma robusta* could not be found in understorey environments comparable with those of the other species, so *C. robusta* was sampled only in gap environments.

Leaf lifetimes were estimated by following survival of leaves over a 12-month period. The height of the principal axis of each juvenile was measured to the apex, and all fully-expanded leaves on this axis were counted. Twelve months later, each plant was revisited, its height remeasured, and survival of leaves recorded. Abscission scars were counted to determine mortality of new leaves initiated after the start of the study period; this was important for species that turn over most or all of their foliage in a single year, such as *Aristotelia serrata*.

Leaf lifetime (years) was estimated as:

$$\frac{n^i}{\left(n^i - n^f\right) + m^n}$$

where $n_i$ = initial number of leaves, $n_f$ = final number surviving from $n_i$, and $m_n$ = mortality of new leaves initiated since the first census (*King, 1994*; *Lusk et al., 2008a*). A few of the marked plants either suffered major damage by herbivores or disturbance during the study, or could not be relocated, and leaf lifespan estimates were eventually obtained from four to five plants of each species in each light environment.

After leaf lifespan measurements, leaves were taken from each of the same plants, for measurement of sun- and shade-leaf structural traits. Depending on leaf size, one
**Table 1 Study species and minimum light requirements of their saplings, estimated as the 10th percentile of the natural distribution of saplings in relation to canopy openness in temperate evergreen forest, Lake Okataina Scenic Reserve, New Zealand.** N indicates the number of sampling points used to compute the light requirements of each species.

| Species | Code | Family | Typical final height (m) | Minimum light requirements (% canopy openness) |
| --- | --- | --- | --- | --- |
| *Beilschmiedia tawa* | Beitaw | Lauraceae | 30 | 0.8 ($n = 72$) |
| *Litsea calicaris* | Litcal | Lauraceae | 30 | 1.3 ($n = 43$) |
| *Hedycarya arborea* | Hedarb | Monimiaceae | 12 | 1.1 ($n = 84$) |
| *Laurelia novae-zelandiae* | Launov | Atherospermataceae | 35 | 1.4 ($n = 92$) |
| *Knightia excelsa* | Kniexc | Proteaceae | 35 | 1.1 ($n = 105$) |
| *Aristotelia serrata* | Ariser | Elaeocarpaceae | 10 | 3.7 ($n = 23$) |
| *Elaeocarpus dentatus* | Eladen | Elaeocarpaceae | 20 | 1.7 ($n = 29$) |
| *Weinmannia racemosa* | Weirac | Cunoniaceae | 25 | 1.7 ($n = 27$) |
| *Alectryon excelsus* | Aleexc | Sapindaceae | 20 | 1.5 ($n = 20$) |
| *Melicytus ramiflorus* | Melram | Violaceae | 10 | 1.9 ($n = 50$) |
| *Myrsine australis* | Myraus | Primulaceae | 6 | 1.8 ($n = 28$) |
| *Geniostoma ligustrifolium* | Genlig | Loganiaceae | 3 | 1.9 ($n = 53$) |
| *Coprosma grandifolia* | Copgra | Rubiaceae | 6 | 1.6 ($n = 104$) |
| *Coprosma robusta* | Coprob | Rubiaceae | 6 | 4.2 ($n = 29$) |
| *Brachyglottis repanda* | Brarep | Asteraceae | 6 | 3.3 ($n = 15$) |
| *Carpodetus serratus* | Carser | Rousseaceae | 10 | 2.6 ($n = 62$) |
| *Pseudopanax arboreus* | Psearb | Araliaceae | 8 | 2.2 ($n = 18$) |
| *Schefflera digitata* | Schdig | Araliaceae | 8 | 1.9 ($n = 35$) |
| *Pittosporum tenuifolium* | Pitten | Pittosporaceae | 8 | 3.5 ($n = 19$) |

to 10 of the youngest fully-expanded intact leaves were taken from each plant, avoiding leaves damaged by herbivores. Leaves were placed immediately in re-sealable plastic bags with moist tissue paper, and fresh weight determined within six hours of removal. Fresh leaves were photographed, and area calculated using ImageJ software (*Schneider, Rasband & Eliceiri, 2012*). Leaf volume was estimated using Archimedes' principle. The leaf was immersed in a small container of water placed on an electronic balance, and displaced volume determined from the change in apparent weight. A small amount of detergent was added to the water to reduce hydrophobicity of leaf surfaces, and reduce bubble formation. Leaf thickness was later estimated dividing volume by area. Leaves were oven-dried at 60 °C for three days before measuring dry weight; drying was initiated within eight hours of leaf excision in all cases, minimizing the effect of dark respiration on non-structural carbohydrates, which can contribute up to 25% of leaf dry mass (*Lusk & Piper, 2007*). Leaf density was then calculated as dry mass / fresh volume. Leaf samples were pooled to obtain one estimate of total nitrogen content of each species in each light environment, using the Dumas combustion method.

**Table 2  Mean leaf trait values (± 1 SD) of temperate evergreen saplings growing in gap and understorey environments, Lake Okataina Scenic Reserve, New Zealand.** Species codes are given in Table 1. nd, no data.

| Species | Light | Canopy openness (%) above trait measurements | Dry matter content (%) | Leaf mass per area (g m$^{-2}$) | Density (g cm$^{-3}$) | Thickness (mm) | Life lifespan (yr) | N (%) |
|---|---|---|---|---|---|---|---|---|
| Aleexc | Gap | 12.2–20.8 | 46.7 ± 3.3 | 88.7 ± 8.4 | 0.45 ± 0.08 | 0.21 ± 0.02 | 2.4 ± 2.0 | 2.2 |
| Aleexc | Shade | 1.3–2.6 | 43.1 ± 3.5 | 55.6 ± 3.3 | 0.49 ± 0.06 | 0.13 ± 0.02 | 3.5 ± 1.5 | 2.2 |
| Ariser | Gap | 13.0–23.1 | 29.2 ± 3.1 | 50.2 ± 27.8 | 0.22 ± 0.03 | 0.24 ± 0.06 | 0.6 ± 0.1 | 2.2 |
| Ariser | Shade | 2.1–2.6 | 25.2 ± 2.7 | 33.8 ± 5.8 | 0.23 ± 0.02 | 0.15 ± 0.02 | 0.7 ± 0.0 | 2.8 |
| Beitaw | Gap | 12.8–22.2 | 42.8 ± 1.1 | 100.0 ± 22.9 | 0.43 ± 0.04 | 0.23 ± 0.03 | 2.3 ± 1.4 | 1.6 |
| Beitaw | Shade | 1.3–2.4 | 40.7 ± 3.3 | 74.4 ± 5.2 | 0.44 ± 0.04 | 0.17 ± 0.00 | 3.8 ± 1.5 | 1.5 |
| Brarep | Gap | 13.4–23.7 | 30.1 ± 2.7 | 83.7 ± 7.4 | 0.29 ± 0.07 | 0.31 ± 0.07 | 1.1 ± 0.1 | 1.5 |
| Brarep | Shade | 1.5–2.8 | 22.1 ± 2.4 | 45.7 ± 10.5 | 0.20 ± 0.03 | 0.23 ± 0.01 | 1.4 ± 0.2 | 1.9 |
| Carser | Gap | 11.6–22.8 | 29.9 ± 4.5 | 59.1 ± 14.8 | 0.25 ± 0.06 | 0.23 ± 0.03 | 1.1 ± 0.5 | 2.2 |
| Carser | Shade | 1.5–2.6 | 24.6 ± 3.7 | 29.6 ± 1.9 | 0.26 ± 0.03 | 0.11 ± 0.02 | 2.2 ± 0.7 | 2.2 |
| Copgra | Gap | 12.2–18.8 | 24.8 ± 4.8 | 68.7 ± 10.2 | 0.20 ± 0.01 | 0.35 ± 0.04 | 1.4 ± 0.3 | 2.2 |
| Copgra | Shade | 1.2–2.3 | 21.5 ± 1.5 | 47.6 ± 16.5 | 0.17 ± 0.03 | 0.28 ± 0.04 | 2.5 ± 0.9 | 1.9 |
| Coprob | Gap | 12.0–28.8 | 27.4 ± 5.4 | 71.5 ± 12.3 | 0.23 ± 0.02 | 0.32 ± 0.05 | 1.1 ± 0.3 | 1.7 |
| Coprob | Shade | nd | nd | nd | nd | nd | nd | nd |
| Eladen | Gap | 13.0 –19.3 | 39.1 ± 6.6 | 78.2 ± 14.8 | 0.32 ± 0.03 | 0.24 ± 0.04 | 1.5 ± 0.3 | 1.5 |
| Eladen | Shade | 1.3–2.6 | 35.6 ± 3.5 | 42.4 ± 2.0 | 0.28 ± 0.04 | 0.15 ± 0.02 | 2.2 ± 1.1 | 1.6 |
| Genlig | Shade | 1.4–2.9 | 15.2 ± 1.8 | 33.4 ± 2.7 | 0.12 ± 0.02 | 0.28 ± 0.04 | 1.7 ± 0.6 | 1.9 |
| Genrup | Gap | 12.8–21.1 | 20.1 ± 2.2 | 64.3 ± 6.2 | 0.18 ± 0.02 | 0.35 ± 0.07 | 1 ± 0.2 | 1.6 |
| Hedarb | Gap | 12.2–18.2 | 24.0 ± 2 | 71.6 ± 25.2 | 0.19 ± 0.04 | 0.37 ± 0.1 | 2 ± 0.6 | 2.7 |
| Hedarb | Shade | 1.3–2.5 | 22.4 ± 2.6 | 58.3 ± 5.3 | 0.19 ± 0.03 | 0.31 ± 0.02 | 5.8 ± 2.8 | 2.9 |
| Kniexc | Gap | 12.8 –25.6 | 42.6 ± 1.8 | 106.5 ± 47.1 | 0.31 ± 0.03 | 0.34 ± 0.1 | 3.8 ± 2.2 | 0.9 |
| Kniexc | Shade | 1.3–2.6 | 41.5 ± 2.2 | 75.6 ± 2.9 | 0.37 ± 0.02 | 0.21 ± 0.02 | 7.7 ± 5.4 | 1.3 |
| Launov | Gap | 12.8–18.2 | 25.2 ± 4.3 | 83.5 ± 25.6 | 0.2 ± 0.03 | 0.37 ± 0.06 | 1.9 ± 0.4 | 1.8 |
| Launov | Shade | 1.3–2.8 | 25.0 ± 3 | 56.6 ± 11.7 | 0.19 ± 0.02 | 0.3 ± 0.05 | 4.6 ± 1.5 | 2.4 |
| Litcal | Gap | 12.8–18.2 | 32.6 ± 5.3 | 83.1 ± 16.2 | 0.28 ± 0.06 | 0.26 ± 0.04 | 2.3 ± 0.9 | 1.6 |
| Litcal | Shade | 1.3–2.9 | 27.8 ± 3.4 | 59.8 ± 8.5 | 0.27 ± 0.04 | 0.22 ± 0.01 | 3.9 ± 1.8 | 1.8 |
| Melram | Gap | 12.2–25.6 | 26.6 ± 4.9 | 73.4 ± 24.4 | 0.25 ± 0.04 | 0.29 ± 0.05 | 1.1 ± 0.2 | 2.4 |
| Melram | Shade | 1.4–2.6 | 22.1 ± 2.5 | 45.0 ± 5.0 | 0.19 ± 0.01 | 0.24 ± 0.02 | 1.7 ± 0.5 | 2.7 |
| Myraus | Gap | 12.5–27.8 | 33.8 ± 0.9 | 83.7 ± 1.0 | 0.28 ± 0.03 | 0.3 ± 0.02 | 1.6 ± 0.4 | 1.2 |
| Myraus | Shade | 1.6–2.8 | 30.3 ± 2.2 | 56.0 ± 2.7 | 0.25 ± 0.04 | 0.23 ± 0.05 | 2.5 ± 0.9 | 1.2 |
| Pitten | Gap | 12.8 –27.5 | 37.0 ± 6.7 | 75.2 ± 27.7 | 0.36 ± 0.05 | 0.21 ± 0.05 | 1.6 ± 0.6 | 1.7 |
| Pitten | Shade | 1.5–2.9 | 29.6 ± 5.3 | 41.7 ± 5.0 | 0.31 ± 0.04 | 0.14 ± 0.02 | 1.8 ± 0.7 | 1.8 |
| Psearb | Gap | 14.0 –25.2 | 32.2 ± 1.2 | 113.8 ± 25.8 | 0.24 ± 0.03 | 0.49 ± 0.05 | 1.3 ± 0.3 | 1.6 |
| Psearb | Shade | 1.4–2.4 | 25.6 ± 2.5 | 65.4 ± 4.4 | 0.19 ± 0.03 | 0.34 ± 0.03 | 2.2 ± 0.5 | 1.6 |
| Schdig | Gap | 13.2–18.8 | 25.4 ± 3.3 | 75.8 ± 8.2 | 0.25 ± 0.01 | 0.3 ± 0.03 | 1.1 ± 0.2 | 2.1 |
| Schdig | Shade | 1.2–2.6 | 18.5 ± 2.0 | 35.2 ± 2.4 | 0.17 ± 0.01 | 0.21 ± 0.01 | 1.9 ± 0.4 | 3 |
| Weirac | Gap | 12.0–24.6 | 36.2 ± 2.3 | 103.3 ± 31.0 | 0.32 ± 0.05 | 0.33 ± 0.05 | 2.7 ± 1.7 | 1.5 |
| Weirac | Shade | 1.3–2.4 | 34.1 ± 3.6 | 71.9 ± 8.3 | 0.32 ± 0.02 | 0.22 ± 0.04 | 3.2 ± 1.3 | 1.3 |

## Quantifying species light requirements

Distributions of juvenile trees 50–200 cm tall were quantified in relation to canopy openness determined from hemispherical photographs. Sampling was carried out on a series of transects run through old-growth stands, including tree-fall gaps of varied sizes and forest margins. Sets of parallel transects were run through forest stands, spaced at least 20 m apart, A total of 748 points were sampled at random intervals (10 to 15 m apart) along transects. Presence of juvenile trees and shrubs 50–200 cm tall was recorded in a circular plot of 1-m diameter, centred on the sample point. Although multiple juveniles of some species were often found in the same plot, only presence or absence data are used in the present analysis. A Nikon Coolpix 4500 digital camera (Nikon, Tokyo, Japan) and an EC-08 fisheye adaptor were used to take a photo at 1.5 m height at each sampling point. A spirit level fitted to the lens cap was used to level the camera, and photos were taken mostly while the solar disc was either obscured by clouds or below the horizon, to avoid errors caused by flaring and reflection. Gap Light Analyzer (*Frazer, Canham & Lertzman, 1999*) was used to estimate % canopy openness from each photo.

The 10th percentile of the distribution of each species in relation to canopy openness was used as an approximation of the lowest light levels tolerated by each species (*Lusk et al., 2008a*). This parameter is referred to hereafter as minimum light requirements (MLR). MLR represents an inversion of traditional shade tolerance ratings, i.e., shade-tolerant taxa have low MLRs, and light-demanders score high. Only species represented on at least 15 sampling plots were considered, yielding 18 species (Table 1).

## Statistical analyses

One-way ANOVA was used to test for interspecific variation in the light environments in which leaf traits were measured. All trait data except leaf dry matter content were log10-transformed before analysis, in order to meet the assumption of additivity of effects (*Quinn & Keough, 2002*). The field sampling procedure meant light environments were effectively nested within species, as species were not all compared in common plots using a full factorial design. Nested ANOVA were therefore used to test for leaf trait differences between sun and shade leaves, and among species. In addition to cross-species correlations, relationships among leaf traits and species light requirements were also measured using phylogenetic least squares regression (PGLS: *Symonds & Blomberg, 2014*), to take into account the influence of phylogeny on trait relationships. COMPARE 4.6 (*Martins, 2004*) was used to carry out PGLS contrasts. A fully-resolved tree of the 19 species was obtained from the Angiosperm Phylogeny Group website (*Stevens, 2001*), although the lack of *Coprosma robusta* in the understorey meant only 18 species were included in the analysis of shade leaf traits. All other analyses were carried out in Statistica (Stat Soft. Inc., Tulsa, OK 74104, USA)

# RESULTS

## Inter- and intraspecific variation in leaf traits

ANOVA showed highly significant interspecific variation in all traits, as well as highly significant effects of light environment on all traits except for leaf density, which was only

**Table 3** Summary of nested ANOVA testing the effects of light environment (gap versus shade) and species on leaf traits ($n = 4-6$ of each species in each light environment). Light environment was nested within species, as species were not all compared in common garden plots. *Coprosma robusta* was omitted from ANOVA, as leaf traits of this species were available only from gaps.

| Effect | SS | df | MS | F | p |
|---|---|---|---|---|---|
| **(a) log(LMA)** | | | | | |
| Intercept | 548.1 | 1 | 548.1 | 67465 | <0.0001 |
| Species | 1.712 | 17 | 0.101 | 12.39 | <0.0001 |
| Light(Species) | 1.805 | 18 | 0.1003 | 12.34 | <0.0001 |
| Error | 1.162 | 143 | 0.0081 | | |
| **(b) log(Density)** | | | | | |
| Intercept | 61.56 | 1 | 61.56 | 6563 | <0.0001 |
| Species | 2.923 | 17 | 0.1719 | 18.33 | <0.0001 |
| Light(Species) | 0.2755 | 18 | 0.01530 | 1.632 | 0.0595 |
| Error | 1.341 | 143 | 0.00938 | | |
| **(c) log(Thickness)** | | | | | |
| Intercept | 64.08 | 1 | 64.08 | 10901 | <0.0001 |
| Species | 2.410 | 17 | 0.1418 | 24.12 | <0.0001 |
| Light(Species) | 1.105 | 18 | 0.06138 | 10.44 | <0.0001 |
| Error | 0.8405 | 143 | 0.00588 | | |
| **(d) Dry matter content** | | | | | |
| Intercept | 154683 | 1 | 154683 | 12414 | <0.0001 |
| Species | 9585.4 | 17 | 563.8 | 45.25 | <0.0001 |
| Light(Species) | 869.8 | 18 | 48.3 | 3.878 | <0.0001 |
| Error | 1781.8 | 143 | 12.5 | | |
| **(e) log(Leaf lifespan)** | | | | | |
| Intercept | 13.04 | 1 | 13.04 | 454.8 | <0.0001 |
| Species | 5.611 | 17 | 0.3300 | 11.51 | <0.0001 |
| Light(Species) | 2.592 | 18 | 0.144 | 5.023 | <0.0001 |
| Error | 3.984 | 139 | 0.02866 | | |

marginally affected by light (Table 3). ANOVA of leaf N data was not possible, due to the lack of replication resulting from pooling of leaf samples.

Interspecific variation of most traits was more marked in shade leaves than in sun leaves. This was especially true of leaf lifespan, which ranged 0.6 to 3.8 years in gaps, and 0.7 to 7.7 years in the understorey (Table 2). Other traits showed narrower ranges of values, spanning 2.4- to 3-fold variation in gaps, and 2.5- to 3.9-fold variation in the understorey.

Leaf lifespan was the trait that responded mostly strongly to light, shade leaves on average living about 70% longer than sun leaves of the same species (Table 2). LMA also responded strongly to light, leaves from plants growing in gaps on average having 58% more dry mass per area than shade leaves of the same species (Table 2); light environment explained just as much variation in LMA as species identity did (Table 3). Light environment had least effect on leaf density and leaf dry matter content, which respectively averaged only 10 and 11% higher in gap plants than in understorey conspecifics; variation in these two traits was

therefore dominated by the effect of species (Table 3). Leaf N of most species was similar in the two light environments (Table 2).

### Trait correlations with species' minimum light requirements

The three traits included in the original leaf economic spectrum (LMA, leaf N, and leaf lifetimes) were tightly coordinated in both sun and shade leaves (Kendall's coefficient of concordance = 0.78 and 0.76 respectively, $P < .0001$; Fig. 1A). They showed varied correlations with the other traits making up the dataset (Table 4).

Species' minimum light requirements ranged from 0.8% (*Beilschmiedia tawa*) to 4.2% (*Coprosma robusta*) (Table 1). Species light requirements were strongly negatively correlated with leaf lifespans, especially those of shade leaves (Table 4; Fig. 1C); leaf lifespan was thus positively correlated with species' shade tolerance. Light requirements were also negatively correlated with LMA (Fig. 1B), although this relationship was only marginally significant in gaps ($P = 0.048$) and non-significant according to PGLS (Table 4). Light requirements were not significantly correlated with any other structural trait or with leaf N, except for a marginally significant correlation with dry matter content of shade leaves under PGLS (Table 4).

## DISCUSSION

Leaf lifespans in the arborescent assemblage at Okataina spanned ~6-fold interspecific variation in gaps, and ~11-fold in the understorey (Table 2). The absence from this temperate assemblage of the fast end of the leaf trait spectrum found in humid tropical forests results in an approximate halving of the log-scaled range of leaf lifespans found in the tropics. Some tropical pioneer trees turn over their foliage in as little as two months (*Poorter & Bongers, 2006*; *Reich et al., 1991*; *Williams, Field & Mooney, 1989*), whereas the shortest-lived leaves found in the assemblage at Okataina were those of *Aristotelia serrata*, which lived about 7 months on average in gaps, and about 8 months in the shade (Table 2). *A. serrata* is a small, fast-growing tree associated mainly with treefall gaps and the edges of tracks at Okataina (*Lusk & Laughlin, 2017*). Woody assemblages in temperate South America appear to span a similar range of leaf economics to that found at Okataina, with no species reported as having a leaf lifespan of <6 months (*Damascos & Prado, 2001*; *Lusk & Corcuera, 2011*; *Lusk et al., 2011b*). However, the few deciduous trees present in both regions probably have slightly shorter leaf lifespans (e.g., *Dungan, Duncan & Whitehead, 2003*). The assemblage at Okataina did not include any species with the very high leaf nitrogen levels found in some tropical pioneers (*Poorter & Bongers, 2006*; *Reich et al., 1991*), although comparable levels (c. 4%) have been reported from some New Zealand native leguminous trees not sampled at Okataina (*McGlone et al., 2004*).

Species light requirements were less consistently related to leaf structural traits than appears to be the case in humid tropical forests, despite strong relationships with leaf lifespan in both gaps and shade (Fig. 1). Although shade-tolerant evergreens in tropical and subtropical rainforests often have dense leaves (*Kitajima & Poorter, 2010*; *Lusk et al., 2010*), light requirements of New Zealand temperate evergreens at Okataina were only weakly (and non-significantly) correlated with density of shade leaves (Table 4), and showed

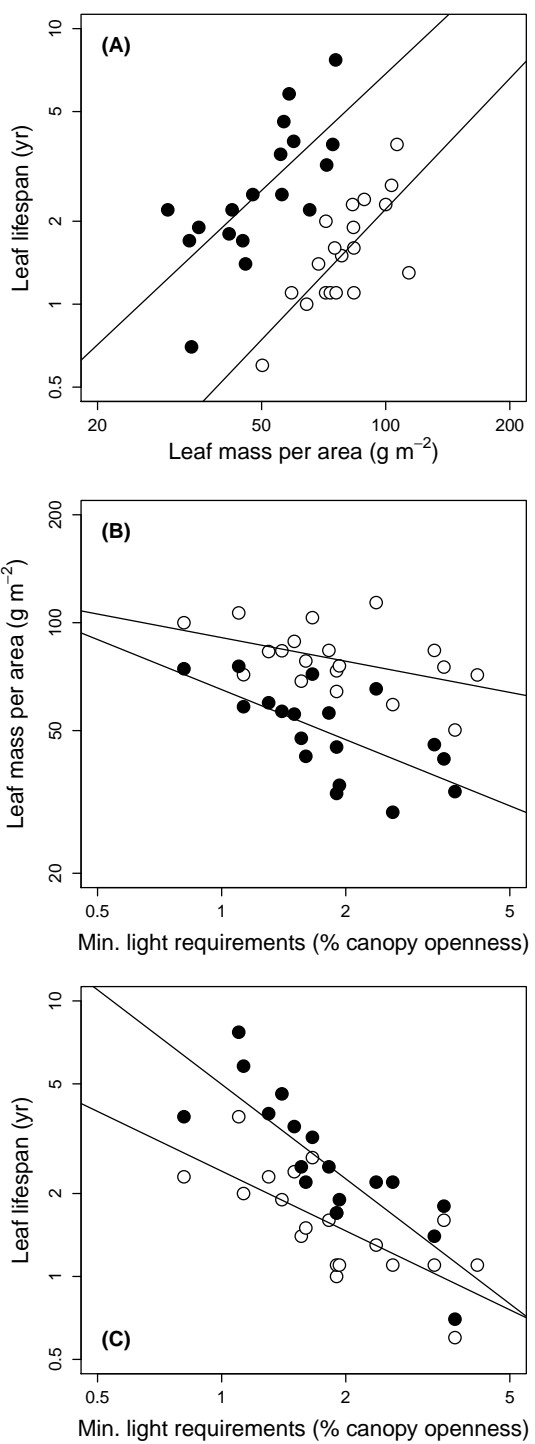

**Figure 1** **Relationships among leaf traits and minimum light requirements of 19 temperate forest ev-ergreens.** Gap and understorey data are shown by open and filled (black) symbols, respectively. (A) Leaf lifespan vs. leaf mass per area; (B) Leaf mass per area vs. species' minimum light requirements; (C) Leaf lifespan vs. species' minimum light requirements. Solid lines show relationships significant at $P = 0.05$; correlation coefficients appear in Table 4.

Table 4 Correlations among sapling light requirements (MLR) and leaf traits of 19 temperate evergreens, New Zealand; only 18 species were sampled in the shade. Values to the lower left of the diagonal show Pearson cross-species correlations; results from phylogenetic least squares regression appear to the upper right.

| | (log)MLR | (log)LMA | (log) Thickness | (log) Density | DMC | (log)N | (log)LL |
|---|---|---|---|---|---|---|---|
| (a) Sun leaves (gaps) | | | | | | | |
| (log)MLR | | −0.41 | 0.00 | −0.24 | −0.26 | 0.12 | −0.73** |
| (log)LMA | −0.47* | | 0.36 | 0.00 | 0.54* | −0.58** | 0.71** |
| (log)Thickness | −0.12 | 0.33 | | −0.59** | 0.00 | −0.15 | 0.07 |
| (log)Density | −0.23 | 0.53* | −0.59** | | 0.93** | 0.00 | 0.52* |
| DMC | −0.30 | 0.57* | −0.48* | 0.93** | | −0.46* | 0.00 |
| (log)N | 0.14 | −0.57* | −0.14 | −0.33 | −0.44 | | −0.50* |
| (log)LL | −0.74** | 0.74** | 0.06 | 0.51* | 0.60** | −0.45 | |
| (b) Shade leaves (understorey) | | | | | | | |
| (log)MLR | | −0.65** | 0.00 | −0.31 | −0.44 | 0.33 | −0.83** |
| (log)LMA | −0.66** | | 0.36 | 0.00 | 0.61** | −0.70** | 0.72** |
| (log)Thickness | −0.24 | 0.34 | | −0.65** | 0.00 | −0.02 | 0.23 |
| (log)Density | −0.32 | 0.49* | −0.65** | | 0.95** | 0.00 | 0.36 |
| DMC | −0.44 | 0.61** | −0.49* | 0.95** | | −0.61** | 0.00 |
| (log)N | 0.23 | −0.55* | −0.02 | −0.43 | −0.55* | | −0.50* |
| (log)LL | −0.84** | 0.73** | 0.22 | 0.38 | 0.48* | −0.30 | |

Notes.
*$P < 0.05$.
**$P < 0.01$.

little relationship with that of sun leaves (Table 4). Notably, leaf densities of two relatively shade-tolerant species (*Laurelia novae-zelandiae* and *Hedycarya arborea*) were among the lowest found at Okataina ($\leq 0.20$ g cm$^{-3}$ in both gaps and shade: Table 2). Species' light requirements were significantly correlated with LMA, especially in the shade (Fig. 1A), in part reflecting the thickness of the leaves of the two aforementioned shade-tolerant species (Table 2). However, the correlation with LMA of sun leaves was weaker when phylogenetic relationships were taken into account by PGLS (Table 4). The long leaf lifespans of *L. novae-zelandiae* and *H. arborea* probably depend to a high degree on chemical (rather than physical) deterrence of herbivores, as both genera are known to be rich in alkaloids and essential oils (*Brophy, Goldsack & Forster, 2005*; *Leitão et al., 1999*; *Urzua & Cassels, 1978*).

Standardization of the light environments in which LMA is measured may be especially important in comparative studies of temperate evergreen assemblages. Not only does interspecific variation of LMA in relation to light requirements run counter to plastic responses to light (as in other evergreen forests: *Lusk et al., 2008b*), but the effect of light on LMA at Okataina was of similar size to that of species identity (Table 3), reflecting a narrower range of the leaf economic spectrum than that present in the arborescent assemblages of humid tropical forests. If this reduced range of leaf traits is typical of humid temperate forest evergreen assemblages, lack of standardization of light environments in some studies may thus explain the variety of reported relationships of LMA with light requirements of temperate evergreen angiosperms, including relationships that are diametrically opposed to that reported here (*Fajardo & Siefert, 2016*; *Hallik, Niinemets &*
*Wright, 2009*). Leaf dry matter content and leaf density were less sensitive than LMA to light environment, species' identity greatly exceeding the effect of light in explaining variation in these traits (Table 3). However, leaf dry matter content and especially leaf density were less useful indicators of shade tolerance than LMA, as they were only weakly related with species' light requirements in the arborescent assemblage at Okataina (Table 4).

The ample representation of small trees with leaf lifetimes of about one year at Okataina (Table 2) suggests the scarcity of the deciduous habit in the New Zealand flora is more a reflection of weak seasonality than of soil fertility (cf. *McGlone et al., 2004*). Deciduous or semi-deciduous species account for <5% of New Zealand's woody flora (*McGlone et al., 2004*); this figure is very low in comparison with temperate floras from continental climates of the northern hemisphere, but falls within the range of values found in other oceanic temperate climates of the southern hemisphere—the proportion of (semi-) deciduous woody species in southern Chile is somewhat higher (c.8%), but only a single deciduous tree is native to Tasmania (*Duretto, 2009*+). Although nutrient conservation is considered one of the advantages of evergreenness (*Aerts, 1995*), those evergreens that replace their entire canopies annually (sometimes termed "leaf-exchangers") will be almost as nutrient-demanding as deciduous trees. Despite moderate soil C:N ratios and low total P at Okataina (*Lusk, Jorgensen & Bellingham, 2015*), six of the 19 study species fit this description, with leaf lifespans of 12–13 months in gaps (Table 3); these species retained their leaves longer in the shade, ranging from a 27% increase in *Brachyglottis repanda* to a 100% increase in *Carpodetus serratus*. Another species (*Aristotelia serrata*) turned over its foliage in well under a year in both gaps and shade (Table 2). All of these seven species are small, fast-growing trees and shrubs that are widespread throughout New Zealand, most of them associated with treefall gaps at Okataina (*Lusk & Laughlin, 2017*),. Elsewhere, another New Zealand *Coprosma* species has also been found to turn over its foliage in about one year (*Richardson et al., 2010*), as has the long-lived canopy tree *Fuscospora fusca* (*Wardle, 1984*).

## CONCLUSIONS

This study makes two main contributions to the literature. Firstly, the dataset reported here—covering over half the arborescent assemblage at the site—confirms that humid temperate forests lack the fast end of the leaf economics spectrum found in their tropical counterparts, reflecting the absence of the fast-growing tall pioneers that exploit treefall gaps at low latitudes (*Lusk, Kooyman & Sendall, 2011a*). Secondly, this study underlines the importance of standardizing the light environments in which leaf traits (especially LMA) of humid temperate evergreens are measured. Without standardization, the strong plastic response of LMA to light may mask interspecific variation in LMA associated with species' light requirements, which is less wide-ranging than in humid tropical forests. Insufficient standardization of light environments, coupled to countergradient variation in LMA (*Conover & Schultz, 1995*; *Lusk et al., 2008b*), may thus explain the lack of consistency in reported relationships of LMA with light requirements of temperate evergreen trees.

## ACKNOWLEDGEMENTS

I thank Tanja Lenz and Teruko Kaneko for technical assistance, and two anonymous reviewers for comments that helped clarify some aspects of the manuscript.

### Funding

This work was funded by the University of Waikato and by the Australian Research Council through Discovery Projects 0878209 and 1094606. The funders had no role in study design, data collection and analysis, decision to publish, or preparation of the manuscript.

### Grant Disclosures

The following grant information was disclosed by the author:
University of Waikato and by the Australian Research Council through Discovery Projects 0878209 and 1094606.

### Competing Interests

The author declares there are no competing interests.

### Author Contributions

- Christopher H. Lusk conceived and designed the research, gathered the data, analyzed the data, contributed reagents/materials/analysis tools, prepared figures and/or tables, authored or reviewed drafts of the paper, approved the final draft.

### Field Study Permissions

The following information was supplied relating to field study approvals (i.e., approving body and any reference numbers):

A research permit to work at the site (66760-RES) was obtained from the Department of Conservation.

### Data Availability

Raw data is available as a Supplemental File.

### Supplemental Information

Supplemental information for this article can be found online at http://dx.doi.org/10.7717/peerj.6855#supplemental-information.

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
