# Peer review of "Leaf functional trait variation in a humid temperate forest, and relationships with juvenile tree light requirements"

_PeerJ, doi:10.7717/peerj.6855_

## Round 0.1 · original submission · Major Revisions

Both reviewers provide extensive comments on this manuscript in an overall positive tone. The study is indeed novel and provides interesting insights into juvenile functional leaf trait variation. There are, however, several places along the manuscript where more clarity and additional citations to support some claims are needed. Also, both reviewers point out to the lack of data on soil fertility and associated limitations of the study in terms of making claims regarding the role of soil nutrients. Reviewer 2 suggest testing, in a simple and direct way the expected changes in LMA that you propose.

Reviewer 1 ·

Basic reporting

No comment

Experimental design

No comment

Validity of the findings

No comment

Additional comments

This manuscript describes a study of naturally occurring juveniles of 19 woody plant species in a temperate rain forest and the relationships between foliar traits (structural and nitrogen concentration) and leaf lifespan in shade and in canopy gaps. Leaf lifespan among species was more strongly correlated with LMA in canopy gaps, while leaf lifespan among species was more strongly correlated with the species’ minimum light requirement in the shade. This is a well written manuscript with clear goals, and well-analysed appropriate data. The study shows the need to understand plasticity in foliar traits. It also shows that the range of leaf lifespan and foliar traits represents a narrower range than in the humid tropics.
Since there are neither data on soil nutrients, no evidence is presented in this manuscript to support a view that seasonality (or lack of it) overrides soil fertility (line 262) as a driver of short leaf lifespan. The data in Lusk et al. (2015, per line 265) is at the whole site scale and may not be applicable for species that are local in their distribution (e.g., lines 127, 226). I think the manuscript could drop lines 260–267, and instead state that the interactions between local soil nutrient availability and light with respect to leaf lifespan and nutrient use efficiency remain to be determined.
Minor points:
Line 55: It seemed an oversight not to include references from New Zealand or Australia. If this is strictly about juvenile leaves, I am unaware of published data, but if interpreted more broadly (cf. line 38; mostly about adult leaves), then consider citing Richardson et al. (2010) and references therein for New Zealand.
Line 56: Remarks about “northern continents” (line 55) are broadly true, but comparisons on leaf lifespans, LMA, etc. can also be drawn with broadleaved evergreen temperate forests of eastern Asia (e.g., Grubb et al. 1975, Kusomoto 1978, Nitta & Ohsawa 1997, Huang et al. 2007) (and subtropical broadleaved forests such as those of Macaronesia). Consider also (cf. line 58) the largely evergreen rain forests comprised of conifers in the Pacific Northwest of North America.
Line 65: Rozendaal et al. (2006) is also very relevant here.
Line 104: Plagianthus regius is the species (for which P. betulinus, not betuloides, is a synonym).
Line 104: McGlone et al. (2004, cited) lists Fuchsia excorticata as fully deciduous, but notes two contrary references about its deciduousness in the northern part of its range; it would be as well to note why this study regards the species as semi-deciduous. McGlone et al. also consider Aristotelia serrata (line 134) as semi-deciduous, which seems to fit the text at line 133 and the data about leaf lifespan in Table 2 (0.7 year length in the shade, 0.6 in the open, also line 224).
Lines 138, 143: What of herbivory by dama wallaby and red deer? (cf. https://www.doc.govt.nz/globalassets/documents/conservation/threats-and-impacts/animal-pests/bay-of-plenty/okataina-wallaby-report.pdf, which lists several of the species in Table 1 as consumed by wallaby or deer).
Line 266: Replace “somewhat” with the mean (± SE) percentage longer in the shade.
Typos:
Line 241: Hyphenate “novae-zelandiae” (as in Table 1)
References
Grubb PJ, Grubb EAA, Miyata I 1975 Leaf structure and function in evergreen trees and shrubs of Japanese warm temperate rain forest I. The structure of the lamina. Botanical Magazine, Tokyo 88, 197–211.
Huang J, Wang X, Yan E 2007 Leaf nutrient concentration, nutrient resorption and litter decomposition in an evergreen broad-leaved forest in eastern China. Forest Ecology and Management 239, 150–158.
Kusomoto T 1978 Photosynthesis and respiration in leaves of main component species. In: Kira T, Ono Y, Hosokawa T (eds) Biological Production in a Warm-Temperate Evergreen Oak Forest of Japan (JIBP Synthesis, Volume 18), pp. 88–98. Tokyo University Press, Tokyo.
Nitta I, Ohsawa M 1997 Leaf dynamics and shoot phenology of eleven warm-temperate evergreen broad-leaved trees near their northern limit in central Japan. Plant Ecology 130, 71–88.
Richardson SJ, Peltzer DA, Allen RB, McGlone MS 2010 Declining soil fertility does not increase leaf lifespan within species: evidence from the Franz Josef chronosequence, New Zealand. New Zeaalnd Journal of Ecology 34, 306–310.
Rozendaal DMA, Hurtado VH, Poorter L 2006 Plasticity in leaf traits of 38 tropical tree species in response to light; relationships with light demand and adult stature. Functional Ecology 20, 207–216.

Reviewer 2 ·

Basic reporting

I have provided a list of suggestions that I hope are found useful to the authors.

Experimental design

I have provided a list of suggestions that I hope are found useful to the authors.

Validity of the findings

I have provided a list of suggestions that I hope are found useful to the authors.

Additional comments

The manuscript Leaf functional trait variation in a humid temperate forest, and relationships with juvenile tree light requirements is an interesting study on the relevance of considering light environments when working on the role of functional traits on plant strategies. However, there are several aspects of the manuscript that were not very clear to me, I hope that my suggestions help and improve this study.


L 49. Can you include a reference to support this statement?
L 50 This statement is confusing, “less well-documented” than? Are the tropics very well-documented? Could you provide a reference to support this statement?
L52-53 This study is a simulation that considers traits of only one (very common) pioneer species from Australia… could you include more references, as this is a rather strong statement and the whole idea does not seem to be supported with literature (I am referring to the previous statements that did not include references).
L53-55 Is this a comparison among evergreens only? If you exclude any deciduous do you still get such a variation?
L61 Hallik et al. 2009 is not on evergreen forests only, this work includes deciduous species and actually reports a correlation between leaf lifespan and shade tolerance.. I am missing the point of this last sentence (L58-61).
L62.
L67 change “than” for that
L67 Lusk et al. 2008b main conclusion is related to thickness and herbivory and this study was made in tropical Australia, how can it support the argument of LMA running counter to inherent variation in shade tolerance?

L 86 “infilled” is written in different font
L109 a space is missing between “(LMA) is”

L112 Indeed thickness does not correlate but can you add a reference that supports that density does? Kitajima´s paper reported the importance of thickness but not density, and again, they (Kitajima and Poorter) highlighted the role of herbivory.
L115 It would be more correct to say “some tropical forests”.. both studies were carried out in the tropics and Poorter 2009 actually includes dry tropics.

L116 (In Table 2 says “Life lifespan” instead of “Leaf”)

L129. Please provide more information on leaf lifespan calculations. How can you calculate leaf lifespan of leaves that live 3 or 5 years by following only 12 months? If I understand you followed “fully expanded leaves” on a branch, but how do you know how old are these leaves? You can tell how many leaves die, but how do you determine the mean and the error of these calculations?

L 142 why depending on leaf size? Why is this relevant?
L 181-187 I understand the value of considering phylogeny, but this is never mentioned in the introduction, so why to do this test? Could you include something in the introduction that could then provide a possible effect of phylogeny on your questions and results? What could we expect if there is a significant phylogenetic signal? How would differences in shade- or sun- traits be?

L194- 205 these statements require statistical support, how do we know that something is “more marked”, that ranges are “narrower”, that a trait responded “more strongly”? The way Table 2 is sorted makes it even harder to figure out the point the author is trying to make here, why not to run a “t test” between sun- and shade- leaves? In that way all this sections would be properly supported.
L199-201 Is this a species-level or community-level comparison? How do you calculate the 59%?

L 209 The usage of bold and bold/underlined I confusing, as the table already has a lot of information, I recommend to change to *0.05 (barely significant) and ** for 0.01 (significant)

L 211-212 But MLR is negatively correlated -0.74 and -0.84 (in bold and underlined) with LL in sun- and shade- leaves respectively, how can you state that this was “especially for shaded leaves? Given this I really do not see support for the statement “leaf lifespan was thus positively (shouldn´t it be negatively) correlated with species’ shade tolerance”

L 213 Light requirements were correlated with LMA in -0.47 (P<0.05) in sun-leaves, but in -0.66 (P<0.01) in shade-leaves, hence, it seems fair to me to say that LMA is actually positively correlated with shade tolerance.

L219-220 These statements seem a bit misleading, as one species is deciduous with very short leaf lifespan, and the 7-year species… without those extremes, variation is actually 4 fold in shaded environments and 3.5 fold in light environments. The 7.7 LL has a standard deviation of 5.4 so do you think that is valid to consider it?

L 220-222 I am sorry but I don´t see the connection between this sentence and the previous one, where is the connection between the tropics and your results.

L 219-234 I am sorry again, but I am bit lost with the point, are you trying to discuss why are there not many deciduous or short-lived leaved species neither in your assemblage or in Chile? Is it not very clear to me and there are studies that have already discuss this, one is Matt´s study, already cited in your paragraph.

L235-245 As previously mentioned, in some aspects of the introduction, in the tropics, thickness has been correlated with herbivory (Kitajima & Poorter 2010) or the study was made in a tropical/diverse region where herbivory may play a relevant role (Lusk et al. 2010). I think that the point you are making here is lacking the relevance, or irrelevance, of herbivory in temperate forests?

L247-255 Very interesting indeed, however, I think that information on soil nutrient availability is lacking, can nutrient availability affect LMA, leaf lifespan or dry matter content? I can see that there is some discussion in the following paragraph, however, soil nutrient availability previous studies also in New Zealand have shown the importance of soil nutrient availability for shade tolerance and plant growth strategies. See Coomes et al. 2009. J. Ecology (/doi.org/10.1111/j.1365-2745.2009.01507.x)


L276-279 I think that this is a very interesting point, but I don´t this is supported with data, temperate humid forests in Australia and Chile have fast-growing tall pioneers, how do you account for that? Or is this an irrelevant point?

L279-282 I may have misunderstood your results, but in order to be sure that I did not, I would like to see t-tests for each species comparing LMA values (gap vs. shade) as I wonder, given the standard deviation values, if indeed there is support for your conclusion.

---

## Round 0.2 · Minor Revisions

This new version addresses most of the initial reviewer´s comments and concerns. The few minor concerns that remain are identified by the reviewer of this new version (e.g. soil fertility).

# "Christopher Lusk has produced a well-written manuscript that evaluates relationships for leaves from a humid temperate forest that are reasonably well known for leaves from humid tropical forests. The relationships are among leaf lifespan, leaf morphology, leaf nitrogen concentrations, the light levels the leaf experienced, and the tolerance of the species for low light levels. Dr Lusk has taken particular care to measure the tolerance of each species for low light levels, and the reported relationships in Figures 1B and 1C are gratifyingly strong. I would say that these figures represent an advance over the humid tropical forest literature where indices of shade tolerance were less precise.

I have only one substantive comment. I am having trouble with the estimate of leaf lifetimes. Estimated leaf lifetimes appear biased downwards. At line 136, leaf lifetimes are Ni / (Ni – Nf + Mn), where Ni is the number of leaves present in an initial census, Nf is the number of survivors from the Ni in a second census one year later, and Mn is the number of leaves that were both born and died between the two censuses. I do not understand why Mn appears in the denominator. The equation Ni / (Ni – Nf) would give mean leaf lifetime if (1) probability of leaf death was a constant independent of leaf age, (2) each leaf cohort present in the initial census had the same number of leaves at its birth, and (3) all leaf cohorts are represented in the numerator and denominator. Conditions 1 and 2 are probably false but, hopefully, not by too much. The equation Ni / (Ni – Nf) is frequently used to estimate leaf lifetimes. But, why is Mn in the denominator at line 136? The cohort that produced these Mn deaths is absent from the numerator. The presence of Mn in the denominator therefore depresses estimated leaf lifetimes. If Mn is included in the denominator, then the numerator should also include all leaves born between the two censuses.

The remainder of my review concerns minor stuff that is easily fixed or that the author can quite reasonably choose to ignore.

Minor stuff the author will want to fix:
Line 56 – fix typo
Line 202 – typo, probably add the word “explained”
Line 209 - A Kendall’s W test appears. Please explain this test and its rationale in the Methods section.
Line 246 – “… weakened by PGLS …” This is not the way to say this. The phylogenetic regression is weaker than the species-level regression would be more appropriate.

Minor stuff for the author to consider and possibly respond to follows:
Methods: Statistical analysis – Should relationships be evaluated using Type II or major axis regression? There is measurement error in all variables. Judging from the relationships in Figure 1, a switch to Type II regression will not change the results at all.
Line 64 and 68 and 251 – Possibly replace the word inherent with the word interspecific. Not at all clear why one source of variation is considered “inherent” implying the other source of variation must not be “inherent”.
Line 178 – Could the author add one line to explain “the assumption of additivity of effects”? I am unfamiliar with this assumption and suspect most readers will be similarly surprised to hear of an ANOVA assumption concerning additivity of effects?
Table 3 – The author might consider adding the percentage of variation explained by species and by light within species as well as the residual variation for each ANOVA. This possible addition would provide evidence up statements like the statement at lines 205 and 254. "

Reviewer 1 ·

Basic reporting

This manuscript is of a high standard four all four criteria for basic reporting.

Experimental design

This manuscript meets all four criteria of experimental design.

Validity of the findings

The manuscript meets all standards set to give confidence in the validity of the findings.

Additional comments

This revised manuscript describes a study of naturally occurring juveniles of 19 woody plant species in a temperate rain forest and the relationships between foliar traits (structural and nitrogen concentration) and leaf lifespan in shade and in canopy gaps. It is a clear, well-written study, generally improved by revision.

In my earlier review, I commented that without data about soil nutrients, I felt that the manuscript did not have sufficient evidence to say that the scarcity of deciduousness in the New Zealand woody flora “is more a reflection of weak seasonality than soil fertility” (line 265 of the current manuscript). In response, I think the author makes a reasonable case that the material should stay. As the author notes in response comments, there is an issue of scale-dependence in assessing whether soil fertility drives leaf lifespan, and I would prefer that this was noted in the manuscript. I would prefer that an additional caveat sentence was inserted in the paragraph beginning at line 264 to say that interactions between local (i.e., within-site, such as Okataina) soil nutrient availability and light with respect to leaf lifespan in the New Zealand woody flora remain to be determined.

Another comment is that the expression of full deciduousness in widespread New Zealand trees is necessarily related to temperature, as shown for Aristotelia serrata (Dungan 2001) (cf. line 104).

Minor corrections
Line 91: insert “the” before “Department”
Line 99: No italics on “Cunoniaceae”
Line 307: Italics on Aristotelia chilensis
Line 393: a 2010 publication cannot be “in press”

Reference
Dungan RJ 2001 Whole‐tree winter leaf‐loss in wineberry (Aristotelia serrata, Elaeocarpaceae) is not related to mean air temperature. New Zealand Journal of Botany 39, 547–550.

---

## Round 0.3 · accepted · Accept

The revised version has adequately addressed most concerns by reviewers. Although there are still some issues that can be contentious, the evidence is not clear and I rather let the readers make their own judgment instead of delaying a paper that by all standards is a nice contribution.

#